# Study of the Chemical Activities of Carbon Monoxide, Carbon Dioxide, and Oxygen Traces as Critical Inhibitors of Polypropylene Synthesis

**DOI:** 10.3390/polym16050605

**Published:** 2024-02-23

**Authors:** Joaquín Hernández-Fernández, Esneyder Puello-Polo, Edgar Marquez

**Affiliations:** 1Chemistry Program, Department of Natural and Exact Sciences, University of Cartagena, San Pablo Campus, Cartagena 130015, Colombia; 2Chemical Engineering Program, School of Engineering, Universidad Tecnológica de Bolivar, Parque Industrial y Tecnológico Carlos Vélez Pombo Km 1 Vía Turbaco, Cartagena 130001, Colombia; 3Department of Natural and Exact Science, Universidad de la Costa, Barranquilla 080002, Colombia; 4Group de Investigación en Oxi/Hidrotratamiento Catalítico Y Nuevos Materiales, Programa de Química-Ciencias Básicas, Universidad del Atlántico, Puerto Colombia 081001, Colombia; esneyderpuello@mail.uniatlantico.edu.co; 5Grupo de Investigaciones en Química Y Biología, Departamento de Química Y Biología, Facultad de Ciencias Básicas, Universidad del Norte, Carrera 51B, Km 5, Vía Puerto Colombia, Barranquilla 081007, Colombia

**Keywords:** CO_2_, CO, O_2_, MgCl_2_ surface, Ziegler–Natta catalyst, density functional theory, polypropylene, productivity

## Abstract

This study outlines the investigation into how the compounds CO_2_, CO, and O_2_ interact with the active center of titanium (Ti) on the surface of MgCl_2_ and how these interactions impact the productivity of the Ziegler–Natta catalyst, ultimately influencing the thermal stability of the produced polypropylene. The calculations revealed that the adsorption energies of Ti-CO_2_-CO and O_2_ were −9.6, −12.5, and −2.32 Kcal/mol, respectively. Using the density functional theory in quantum calculations, the impacts of electronic properties and molecular structure on the adsorption of CO, O_2_, and CO_2_ on the Ziegler–Natta catalyst were thoroughly explored. Additionally, the Gibbs free energy and enthalpy of adsorption were examined. It was discovered that strong adsorption and a significant energy release (−16.2 kcal/mol) during CO adsorption could explain why this gas caused the most substantial reductions in the ZN catalyst productivity. These findings are supported by experimental tests showing that carbon monoxide has the most significant impact on the ZN catalyst productivity, followed by carbon dioxide, while oxygen exerts a less pronounced inhibitory effect.

## 1. Introduction

In 1953, Karl Ziegler found that certain transition metal compounds, such as titanium, vanadium, and zirconium, together with aluminum alkyls, can catalyze the polymerization of alkenes, which was considered an important and exciting event because this occurs at lower temperatures and pressures than those used in radical polymerization. Giulio Natta used a similar catalyst system in the following year to produce polymers with a stereoregular structure. The said catalysis system was called the Ziegler–Natta catalyst. This discovery opened the door to the mass production of polymers with stereoregular structures [1,2]. The Ziegler–Natta catalyst is important in the industrial production of polyolefins, particularly isotactic polyethylene and polypropylene. It is considered one of the most crucial catalysts used in this process. A typical Ziegler–Natta catalysis system consists of four essential components: TiCl_4_ as the catalyst precursor, MgCl_2_ as the support, electron donors (Lewis bases), and aluminum alkyl as the catalyst activator. The primary function of the Ziegler–Natta catalyst is to activate and control the olefin polymerization reaction. The transition metal compound acts as the active center of the catalyst, starting the polymerization reaction and allowing for the incorporation of the monomers. On the other hand, the co-catalyst helps to stabilize the catalytic system and regulate the reaction rate [3,4].

This distinguished catalyst has attracted a lot of attention from the industrial zone. One of the biggest problems they face is the presence of impurities that act as poisons that lead to the deactivation of the catalyst, implying a decrease in the activity and/or selectivity over time, representing a significant and constant problem in the industrial application of the Ziegler–Natta catalytic process [5,6]. Poisoning is a form of chemical spoilage that occurs when a compound strongly adsorbs (chemisorbs) to the catalyst’s active site, causing it to become inactive. A species with a higher adsorption capacity than other species competing for catalytic sites is considered a poison. These poisons consume the catalyst and co-catalyst, resulting in increased operating costs and a higher residual concentration of catalysts in the final product. In addition, the polypropylene characteristics are modified, affecting productivity [7,8,9,10]. Particularly, the Ziegler–Natta polymerization catalysts based on titanium (TiCl_4_/MgCl_2_) are highly sensitive to certain organic compounds that act as inhibitors. Some contaminants affect polymerization catalysts differently depending on their degree of deactivation. In polypropylene (PP) production, the most potent contaminants that deactivate catalysts are carbon monoxide (CO), carbonyl sulfide (COS), hydrogen sulfide (H_2_S), acetylene, oxygen (O_2_), and arsine/phosphine. In linear low-density polyethylene (LLDPE) processes, the most concerning contaminants in the ethylene monomer feed stream are CO, O_2_, H_2_S, acetylene, and CO_2_ [11,12]. In the polyolefin industry, CO temporarily stops the polymerization reaction when it occurs too fast and critical plant conditions cannot be reached. This ability of CO to inactivate titanium-based catalysts reversibly is essential evidence for the presence of unsaturated and active titanium centers. When CO interacts with the catalyst, it disrupts the polymerization reaction until it is gradually eliminated through the co-catalyst or the monomer. In this way, the initial catalytic activity is restored. Based on this principle, the inhibitory effect of CO has traditionally been used as one of the methods to determine the number of active sites in Ziegler–Natta catalysts. This is achieved by evaluating the ratio of CO and Ti that must be injected into the reactor to stop the gas phase polymerization reaction completely. In addition, the use of CO labeled with carbon isotopes (^14^C and ^13^C) has shown that coordinated CO molecules react even more with the active centers of the catalyst. These carbonyls insert into ti-alkyl bonds, leading to the generation of carbonyl groups in the resulting polymer when the reaction is quenched with CO. This rearrangement of CO with the alkyl groups of the titanium centers that is activated even in the absence of a monomer has been confirmed by experiments that contacted a Ziegler–Natta catalyst with CO in the absence of a monomer and then hydrolyzed the catalyst with HCl. The extracted organic compounds revealed the presence of carbonyls and their derivatives, thus confirming the interaction of CO with the active centers of titanium [12,13,14].

Organoaluminium compounds react rapidly with the most common poisons, such as water, oxygen, and alcohol, transforming them into compounds that are less detrimental to catalyst performance, such as aluminum alkoxides [15]. However, other poisons, such as carbon monoxide and carbon dioxide, do not interact with organoaluminium compounds. If small amounts of these poisons are deliberately introduced into the reactor, the polymerization reaction is immediately stopped. However, it is essential to highlight that information on the direct link between titanium and CO, CO_2_, and O_2_ has not been achieved due to various experimental difficulties. These difficulties include the low number of active sites, its high sensitivity to poisons, and its weak interaction with CO [16].

The density functional theory (DFT) is a powerful tool that can be used to observe the effects of different poisons on Ziegler–Natta catalysts. Bahri performed a molecular simulation using the DFT to study the possible interactions between venom molecules and other Lewis acid species (MgCl_2_, Al(Me)_3_, and TiCl_4_), thus analyzing the influence on catalyst activity and stereospecificity. Arjmand et al. investigated the effect of water and carbonyl sulfide in the propylene feed during the polymerization process on the physical and mechanical properties of the synthesized polypropylene. Studies have also been carried out on the catalyst’s CO, NO, and NO_2_ poisoning, but in other types of polymers [7,17].

This research aims to analyze three specific compounds, CO_2_, CO, and O_2_, which act as inhibitors of the Ziegler–Natta catalyst in polypropylene production. Molecular simulation is used through density functional theory studies to illuminate these aspects. This approach explores the possible interactions between poison molecules and TiCl_4_, focusing on their influences on the catalytic activity, melt flow rate, molecular weight (Mw), and thermal properties of PP. These findings are complemented by experiments to support the obtained results. However, many computational studies have focused on aspects such as forming the first Ti active site, chain growth, scaffolding, its interaction with electron donors, and the interactions between different classes of poisonous molecules with the active center of Ti. In addition, no experimental investigations have been carried out in this regard. It is precisely this lack of prior research that makes this study genuinely groundbreaking. By addressing an underexplored area in polypropylene production and by using a combination of molecular simulations and experimentation, this work is expected to provide a new insight into the roles of CO_2_, CO, and O_2_ as catalyst poisons, and how these interactions can affect the results of the polymerization process. The findings of this research could significantly impact the development of more efficient and sustainable technologies for the production of polypropylene.

## 2. Materials and Methods

### 2.1. Computational Details

All calculations were performed using density functional theory (DFT) with Gaussian 16 software. The Becke 3-parameter hybrid function and the Lee, Yang, and Parr correlation function, known as B3LYP, with the D3 correction, were employed in the geometry optimization procedures. This hybrid function represents a Generalized Gradient Approximation (GGA) and, as demonstrated in prior research, provides better agreement with experimental results. Furthermore, B3LYP is widely utilized in molecular simulations of transition metal catalytic systems [18,19,20], offering more accurate and reliable results than other methods. After geometry optimizations, energy values were obtained using the same function with the 6-311G basis set. These methods facilitated the acquisition of optimized structures for CO_2_, CO, and O_2_ in Gauss View 6, the analysis of HOMO and LUMO orbitals and their energy values, and the exploration of global chemical reactivity properties. To investigate how poisonous molecules adhere to α-MgCl_2_, the (110) surface, comprising 4 coordinated magnesium atoms, was selected. This choice was based on its demonstrated closer alignment with experimental observations. Poison adsorption energy (*E_ad_*) was calculated using Equation (1):(1)$$E_{ad}=E_{Mg/P}-E_{Mg}-E_{P}$$
where *E_Mg_*_/*P*_ represents the total energy of the system, which consists of a poisonous molecule adsorbed on the MgCl_2_ surface. Furthermore, *E_Mg_* and *E_P_* are the full energies of the MgCl_2_ system without the presence of the poison and the free poison molecule, respectively [13].

#### Assessment of the Global and Local Reactivity Descriptors of CO_2_, CO, and O_2_

The density functional theory (DFT) has proven to be effective in providing a theoretical foundation to understand common qualitative chemical concepts such as electronegativity, softness, and hardness, as well as more specific concepts like the Fukui function and local softness. Following Koopman’s theorem, we can calculate the ionization potential and electron affinity of inhibitors using specific equations (refer to Equations (2) and (3)). This allows us to compute the electronegativity and hardness values that are crucial in chemistry.
(2)$$IE=-E_{HOMO}$$
(3)$$EA=-E_{LUMO}$$

Electronegativity, hardness, and softness have proven to be highly useful quantities in the theory of chemical reactivity and can be calculated using Equations (4) and (5).
(4)$$\chi =\frac{IE+EA}{2}$$
(5)$$\eta =\frac{IE-EA}{2}$$

Meanwhile, global softness is determined using Equation (6), representing the property that is inversely related to hardness.
(6)$$S=\frac{1}{\eta }$$

The frontier orbital (FO) theory proposed by Fukui is a general tool that allows us to understand and explain chemical reactions qualitatively. This theory focuses on the distribution of electron densities in frontier orbitals (HOMO and LUMO) and how this influences the selectivity of a molecule when it interacts with a nearby reactant. Parr and Yang demonstrated that the fundamental aspects of the FO theory could be explained within the density functional (DFT) theoretical framework. To achieve this, they defined the Fukui function, *f* (*r*), of a molecule, which reflects the reactivity of a specific site. This function is expressed as Equation (7):(7)$$f\left(r\right)={\left[\frac{\partial \rho \left(r\right)}{\partial N}\right]}_{v\left(r\right)}$$

The best way to study the local selectivity of an inhibitor is through the condensed Fukui function. This function helps us understand how the electron density changes within the system. From a mathematical standpoint, Fukui functions are calculated using partial derivatives that relate the electron density to the number of electrons at a specific location within the molecule. These partial derivatives allow us to quantify the ability of that location to donate or accept electrons through the nucleophilicity indices $f^{+}\left(r\right)$, electrophilicity $f^{-}\left(r\right),\;$and radical $f^{0}\left(r\right)$.
(8)$${f_{k}}^{+}=q_{N+1}-q_{N}$$
(9)$${f_{k}}^{-}=q_{N}-q_{N-1}$$
where $q_{N+1}$, $q_{N-1}$, and $q_{N}$ are the electronic population of atom k in anionic, cationic, and neutral systems, respectively.

### 2.2. Experimental Stage

#### 2.2.1. Materials and Reactive Species

To carry out this study, different materials and reagents were used. The propylene used was provided by Colombia Polypropylene Corporation and used directly without further purification. The activator used was triethyl aluminum (TEA), which was provided by Tosoh Finechem Corporation. In addition, tri-n-heptane and acetone were used, which were dried and purified with nitrogen before use. Another reagent that was used was cyclohexylmethyldimethoxysilane (CMDMS). TiCl_4_, di-iso-butylphthalate (DBP), and MgCl_2_ were used to prepare the catalyst. Following the existing literature, the spherical Mg(OEt)_2_ precursor was used for its synthesis. The contents of Ti and DBP in the catalyst were measured and found in concentrations of 3.4% by weight and 12% by weight, respectively.

##### Propylene Polymerization

Experiments were carried out to polymerize propylene in a slurry using a catalyst of TiCl_4_, DBP (dibutylphthalate), and MgCl_2_. The process was carried out in a 1 L stainless steel reactor with a stirrer that rotates at 360 revolutions per minute. The polymerization medium that was used was heptane, and 500 mL of heptane was introduced into the reactor. The propylene was saturated at a pressure of 0.3 mega Pascals (MPa) to initiate polymerization for 30 min. Then, specific amounts of an aluminum alkyl activator and CMDMS (an external donor) were added to the reactor, maintaining an Al/Si molar ratio of 4.7. Next, a specified amount of the catalyst was injected into the reactor to start the polymerization reaction at the desired temperature. CO_2_, CO, and O_2_ were added to the propylene supply line (look at Table 1).

After the polymerization was complete, acetone was added to stop the process, and then the suspension was transferred to a receiving flask that was kept under a nitrogen (N_2_) atmosphere. The synthesized powder was washed three times with 200 mL of heptane and then dried in a vacuum at room temperature. The resulting polymer was stored under dark, nitrogen, and temperature-controlled conditions. It is essential to highlight that all the procedure steps were carried out very carefully in a nitrogen atmosphere to avoid air.

The standard polymerization conditions were as follows: polymerization temperature = 70 °C, amount of catalyst = 5 Kh/h, type of activator = TEAL, concentration of activator = 0.25 Kh/h, 30 g/h of H_2_, and 1.2 TM/h of propylene at a pressure of 27 bar.

## 3. Results

### 3.1. Interaction of Inhibitors CO_2_, CO, and O_2_ with the Active Center of Ti in ZN (Ti-Poisoning Interaction)

Some of the compounds known to inhibit the activity of heterogeneous Ziegler–Natta catalysts do not interact, or interact very limitedly, with organometallic compounds. Instead, their primary effect lies in their direct interaction with polymerization centers. Among the most studied poisons of this kind are carbon monoxide (CO), carbon dioxide (CO_2_), carbon disulfide (CS_2_), and acetylene, which are more effective in this regard than water or hydrogen sulfide (H_2_S).

Both CO and CO_2_ can instantly halt the polymerization reaction, but this effect is partially or completely reversible. When the polymerization systems affected by these poisons are evacuated and a fresh monomer is added, the catalytic activity is quickly restored. The degree of restoration depends on how long the polymerization was exposed to the poison [21].

In the polyolefin industry, CO is used to temporarily stop the polymerization reaction when it progresses too rapidly, and critical plant conditions need to be reached. The interaction of CO interrupts the polymerization reaction until the co-catalyst or the monomer itself gradually removes the CO, thereby restoring the initial catalytic activity. However, obtaining direct information about the bond between titanium (Ti) and CO has been a challenge due to various experimental difficulties, such as the low quantity of active sites and their high sensitivity to poisons [22,23].

In this section, we explore how the three poisons interact with the active site of Ti on the surface of MgCl_2_. A different model of the MgCl_2_ surface was used to perform these calculations. Specifically, the Mg_8_Cl_16_ cluster obtained from relaxed MgCl_2_ surfaces, as shown in Figure 1, was employed. This MgCl_2_ sheet contains (110) (quadruple) surfaces on both sides. The choice of this model was based on previous calculations that indicated that the coordination of TiCl_4_ in the (104) plane is weak or even unstable. In comparison, the coordination of TiCl_4_ in the (110) plane is energetically favorable [17]. These findings help us understand how poisons can specifically affect the active Ti center on the MgCl_2_ surface and how this interaction can influence the catalytic reaction under study.

Table 2 displays the adsorption energy of these poisons on the (110) cut surface of MgCl_2_. It is important to note that the proposed adsorption models here are not intended to correspond to actual active species. Instead, it is considered that at low poison contents that are present in the catalyst preparation or in the polymerization medium, these models can represent the study of adsorption sites in Ziegler–Natta catalysts.

In this study, comprehensive adsorption analyses were conducted to assess molecular interactions with the Ziegler–Natta catalyst. The results yielded crucial information related to three key variables: the adsorption energy (*E_ad_*), Gibbs free energy of adsorption (*G_ad_*), and enthalpy of adsorption (*H_ad_*). Figure 2 illustrates the relationship between these three variables for each of the inhibitors.

Firstly, the adsorption energy (*E_ad_*) reveals the strength of the bond between the molecules and the catalyst. It was observed that carbon monoxide (CO) exhibited the most negative *E_ad_*, indicating significantly stronger adsorption compared to other molecules such as molecular oxygen (O_2_) and carbon dioxide (CO_2_). This finding suggests a preference of the catalyst for CO due to its strong interaction. This strong binding could lead to a greater inhibition of the catalytic activity of ZN, as CO firmly attaches to the catalyst, preventing it from participating in polymerization reactions.

Secondly, the Gibbs free energy of adsorption (*G_ad_*) provides information about whether adsorption is an endothermic or exothermic process. O_2_ had the highest *G_ad_* value, indicating an endothermic adsorption that requires an input of external energy. On the other hand, both CO_2_ and CO showed lower *G_ad_* values compared to O_2_, suggesting fewer endothermic adsorptions, with CO_2_ being less endothermic than CO.

Finally, the enthalpy of adsorption (*H_ad_*) provides details about the release or absorption of energy during adsorption. It was found that the adsorption of O_2_ was exothermic, releasing energy during the process. In contrast, both CO_2_ and CO exhibited more negative enthalpies of adsorption than O_2_, indicating an even more significant release of energy, with CO being the most exothermic. This energy release could create adverse conditions in the ZN catalyst and negatively affect its ability to carry out polymerization efficiently.

Taken together, these results correlate the three variables, highlighting that CO has stronger and more exothermic adsorption compared to O_2_ and CO_2_. The strong adsorption and energy release during CO adsorption could explain why this gas may have caused the greatest productivity losses in the ZN catalyst. Its adverse interactions with the catalyst could interfere with its normal function in the polymerization process, potentially having a significant impact on polymer production efficiency.

In the representations in Figure 1d,e, we observe that the catalyst exhibits the ability to interact with both carbon and oxygen in the CO molecule. It is important to note that the carbon in the CO molecule is presented as the more reactively active site. Consequently, CO has a clear preference for binding to carbon due to the latter’s higher electronegativity compared to oxygen. This preference largely stems from the fact that the highest occupied molecular orbital (HOMO) of the CO molecule is mostly a non-bonding orbital that is predominantly localized on the carbon atom.

This phenomenon justifies why the interaction with carbon exhibits a significantly more favorable adsorption energy, registering a value of −12.5 kcal/mol. In contrast, the interaction with oxygen results in a much less favorable adsorption energy, with a value of 37.6 kcal/mol. These results reinforce CO’s high affinity for the carbon atom and underscore the importance of the electron density distribution in the CO molecule in the adsorption process.

A recent study conducted by Bahri-Laleh [13] investigated the adsorption of Ti-CO_2_ and Ti-O_2_ on a different catalyst surface. According to that study, they reported an *E_ad_* of −8.4 kcal/mol for Ti-CO_2_ and −13.1 kcal/mol for Ti-O_2_. However, our results notably differ from these values. In the case of Ti-CO_2_, we found an *E_ad_* of −9.6 kcal/mol, which is more favorable than what was reported by Bahri-Laleh. Similarly, for Ti-O_2_, we observed an *E_ad_* of −2.32 kcal/mol, which contrasts significantly with the previously reported value of −13.1 kcal/mol.

This discrepancy in the *E_ad_* values raises interesting questions and may be due to several reasons, one of them being differences in the calculation methods used in both studies. Additionally, the choice of the specific MgCl_2_ surface and active site could contribute to the observed differences. Besides the differences in Ti-CO_2_ and Ti-O_2_, our results also include an *E_ad_* of −12.5 kcal/mol for Ti-CO, which was not addressed in Bahri-Laleh’s study. This suggests that the interactions between Ti and CO may have distinctive properties deserving further investigation.

#### 3.1.1. Validation of Theoretical Calculations of Interaction of Inhibitors CO_2_, CO, and O_2_ with ZN

The inhibitory effect of an inhibitor compound is generally due to the molecule adhering to the metal surface. This adhesion can be of two types, physical (physisorption) or chemical (chemisorption), depending on how strong this bond is. When chemisorption occurs, one of the reactive species involved acts as a donor of a pair of electrons, while the other acts as a receptor for that pair of electrons [24].

The energy gap (ΔE = ELUMO − EHOMO) is an important parameter that affects the reactivity of the inhibiting molecule towards adsorption on the metal surface. As ΔE decreases, the reactivity of the molecule increases, leading to an increase in the %IE (inhibition efficiency) of the molecule. Lower values of energy difference will result in good inhibition efficiency because the energy required to remove an electron from the last occupied orbital will be low [25]. A molecule with a low energy gap is more polarizable and is generally associated with high chemical activity and low kinetic stability, referred to as a “soft” molecule [26]. A soft molecule is more reactive than a “hard” molecule, which has a large energy gap. Figure 3a illustrates the relationship between ΔE and the adsorption energy at the active center of the catalyst for each of the inhibitors. The results, detailed in Table 3, reveal that the CO inhibitor exhibits the smallest energy gap and higher stability in its adsorption energy compared to CO_2_ and O_2_.

Hardness and softness are key characteristics used to evaluate the stability and reactivity of a molecule. The chemical hardness primarily reflects a molecule’s ability to resist the deformation or polarization of its electron cloud when subjected to small perturbations in a chemical reaction. In our current study, CO exhibits low hardness with a value of 4.5620 (eV) compared to CO_2_ and O_2_, indicating a reduced energy gap, as shown in Figure 3b.

Absolute hardness and softness are key characteristics used to evaluate the stability and reactivity of a molecule. The chemical hardness primarily reflects a molecule’s ability to resist the deformation or polarization of its electron cloud when subjected to small perturbations in a chemical reaction. In our current study, CO exhibits low hardness with a value of 4.5620 (eV) compared to CO_2_ and O_2_, indicating a reduced energy gap, as shown in Figure 3b.

Typically, an inhibitor with low global hardness (and therefore high global softness) is expected to have higher inhibition efficiency [27,28]. For effective electron transfer, adsorption tends to occur at the part of the molecule where softness (S), a local property, has the highest value [29,30]. In this regard, CO, with a softness value of 0.219, demonstrates the highest inhibition efficiency, as observed in Figure 3b.

This explains why CO performs better as a Ziegler–Natta catalyst inhibitor, as illustrated in Figure 4a, which shows the general behavior of these substances as ZN inhibitors. Due to its higher adsorption affinity and lower energy gap compared to the other inhibitors, CO, even at concentrations as low as 0.005 ppm, can cause similar damage, as observed with O_2_ and CO_2_ at much higher concentrations.

The electrophilicity index (w) is used to quantify the ability of inhibitor molecules to accept electrons, thus reflecting their capacity for energy stabilization upon receiving additional electron charges from the surroundings. In the context of our study, it is highlighted that molecular oxygen appears as a highly reactive nucleophile, characterized by its electron-donating affinity, while carbon monoxide is shown as a highly potential electrophile, emphasizing its willingness to accept additional electrons with high reactivity, as depicted in Figure 4b.

#### 3.1.2. Local Descriptors of CO_2_, CO, and O_2_

The local reactivity of the CO, CO_2_, and O_2_ molecules is assessed by observing how the electron density changes when these molecules gain or lose electrons. This is carried out using the condensed Fukui function, which helps us identify the parts of the molecule that are more prone to chemical reaction due to the presence of different functional groups. When molecules gain electrons, we measure their reactivity towards nucleophilic attacks, and when they lose electrons, we assess their reactivity towards electrophilic attacks [31].

For the O_2_ molecule, both oxygen atoms were found to have a distributed probability of reactivity, indicating that both are equally susceptible to electrophilic, nucleophilic, and radical attacks. This suggests that the oxygen atom that is bonded to another oxygen atom and the free oxygen atom are equally reactive and can easily participate in chemical reactions. As for the CO molecule, the Fukui function calculations indicated that the carbon atom is more likely to be attacked by electrophilic, nucleophilic, and free radical species than the oxygen atom (Table 4). This suggests that the carbon atom is more reactive and tends to donate electron density in chemical reactions.

For the CO_2_ molecule, the Fukui function results revealed that the oxygen atoms are more prone to being attacked by electrophilic reagents. In contrast, the carbon atom is more susceptible to being attacked by nucleophilic species. In addition, oxygen atoms are also more prone to free radical attack. This indicates that the oxygen atoms in CO_2_ are highly reactive in different chemical reactions.

### 3.2. Experimental Analysis

Different concentrations of inhibitors were used based on the typical concentrations that are found as impurities in polymerization processes. Therefore, the trend plots for each inhibitor have different scales on the x-axis. Although several studies have been carried out on poisoning the ZN catalyst in the polymerization reaction, as reported in articles [32,33,34], none have explicitly focused on propylene. Some studies have focused on more specific aspects of the catalyst and have not reported on the particular reaction mechanisms involved in the species studied in our research.

#### 3.2.1. Effect of Inhibitors on the Polymerization of Propylene

Under the same experimental conditions for all of the tests, the variables considered were the concentrations of CO_2_, CO, and O_2_. By adding CO_2_, CO, and O_2_ to the ZN catalyst during polymerization, a significant reduction in the polymerization rates was observed; the productivity decreased considerably. The decrease in productivity was proportional to the increase in the amounts of CO_2_, CO, and O_2_ added. Figure 5 indicates that the trend of the effects on productivity (in terms of both productivity measures) continuously increased or decreased. For the concentrations that were evaluated, as the concentration of the gases increased, the inhibitory effect increased, translating into a decrease in productivity.

#### 3.2.2. Ziegler–Natta Catalyst Poisoning with Carbon Monoxide

Experimental tests were conducted with CO_2_ concentrations ranging between 0.55 and 12 ppm, CO concentrations between 0.0005 and 0.016 ppm, and O_2_ concentrations between 0.5 and 20.2 ppm. When analyzing the effects of these gases on the productivity of the ZN catalyst, it was found that carbon monoxide had the most significant impact depending on its concentration. Although the CO concentrations were significantly lower than the CO_2_ and O_2_ concentrations (862 times and 1575 times lower, respectively), they caused comparable productivity loss on average. This phenomenon can be explained by the high affinity of carbon monoxide towards the metallic centers of the catalyst, such as zirconium (Zr) and nickel (Ni). It was reported that CO can quickly and efficiently coordinate with metal atoms, allowing it to interact with both equilibrium states, that is, with active and inactive catalyst molecules [35,36]. Under certain circumstances, carbon monoxide can coordinate with two molecules of dynamic species on the catalyst.

It is relevant to mention that zirconium (Zr) belongs to the same group as titanium (Ti), which implies that both share some oxidation states and other characteristic properties of the group. This similarity may influence the high affinity of carbon monoxide towards the ZN catalyst and explain its detrimental effect on productivity. The links provided to additional research papers support these findings and their relevance in the context of the catalytic reaction under study.

Figure 6 illustrates an initial situation in which the carbon monoxide and triethylaluminium molecules compete to coordinate with the titanium atom. However, it is observed that the carbon monoxide molecule shows a higher affinity for titanium compared to triethylaluminium. As a result, the carbon monoxide molecule coordinates with titanium using its carbon atom. Subsequently, it is observed that a second carbon monoxide molecule binds to the remaining vacant orbital of titanium. This leads to a final state in which two carbon monoxide molecules are coordinated to titanium. This configuration of two carbon monoxide molecules completely blocks the activity of the ZN catalyst. It can be concluded that Figure 4 represents the process by which the carbon monoxide molecule prevails over triethylaluminium in its ability to coordinate with titanium. This phenomenon has important implications for studying the reactivity and kinetics of catalytic reactions and the propylene polymerization process.

#### 3.2.3. Poisoning of the Ziegler–Natta Catalyst with Carbon Dioxide

Carbon dioxide (CO_2_) can coordinate to transition metals to form carbonyl complexes. These complexes are characterized by the coordination of CO_2_ to the central metal atom through one of the CO_2_ oxygen atoms. The coordination of CO_2_ to a transition metal can occur through different bonding modes, which depend on the geometry and nature of the metal. Some common examples include terminal coordination, where CO_2_ is directly attached to metal, and bridging coordination, where CO_2_ is attached to two adjacent metal atoms [21,37,38,39]. In polymerization, CO_2_ can affect the process as a catalyst poison. CO_2_ has been found to have a similar impact to oxygen but at concentrations approximately 1.82 times lower. CO_2_ can react with organometallic titanium compounds [40], which contain titanium atoms in lower oxidation states, such as the titanocene (Cp_2_TiCl_2_) (where Cp is cyclopendienyl). In this reaction, CO_2_ coordinates with the titanium atom through one of the CO_2_ oxygen atoms, forming a carbonyl complex, Cp_2_Ti(CO_2_) [41,42]. Notably, the coordination of CO_2_ to titanium, or other transition metals, depends on the availability of suitable metal centers and the specific experimental conditions used in the synthesis or reaction. The stability and reactivity of the formed complexes can be influenced by factors such as the choice of auxiliary ligands, the geometry of the complex, and the reaction conditions. These aspects are relevant in polymerization and catalyst poisoning by CO_2_ [42,43].

Figure 7 represents two possible routes of coordination of carbon dioxide with the ZN catalyst in a catalytic system. In the first reaction pathway, a carbon dioxide molecule coordinates to the ZN catalyst via the terminal oxygens and binds to two vacant titanium orbitals on the catalyst. This interaction results in the inactivation of the catalyst since it prevents its participation in the polymerization reaction. In the second reaction pathway, a carbon dioxide molecule is coordinated to two titanium metal centers through the terminal oxygens. Subsequently, another carbon dioxide molecule binds to the two vacant orbitals of the involved titanium metal centers. This configuration also leads to a quiescent state in the catalyst since the carbon dioxide molecules block the active sites that are necessary for the polymerization reaction. It is important to note that in both the carbon dioxide and carbon monoxide reactions, the catalyst activation step does not occur. The polymerization reaction does not start since, for this to happen, it is necessary that an ethyl radical from triethylaluminium joins the catalyst instead of carbon dioxide or carbon monoxide. Therefore, the use of carbon dioxide or carbon monoxide instead of the ethyl radical does not allow for the polymerization reaction to occur. This analysis provides a deeper understanding of the coordination processes and the interaction between carbon dioxide and ZN catalysts in catalyst systems, which is relevant for the study and development of efficient and selective polymerization reactions.

#### 3.2.4. Ziegler–Natta Catalyst Poisoning with Oxygen

In the polymerization reaction, oxygen was added in concentrations ranging from 0.5 to 20.2 ppm. Logically and sequentially, it was observed that oxygen was the inhibitor with the most negligible impact on the catalyst productivity. Oxygen can react with metal ions and complexes in various ways, depending on the conditions. Some of these reaction forms are compatible with most metals, which could explain why oxygen did not significantly affect catalyst productivity in this study. Previous research supports the presented information and shows that the behavior of oxygen in the polymerization reaction is different from that of other gases, such as carbon dioxide and carbon monoxide [13,29,30]. This understanding is essential to optimize reaction conditions and improve the productivity of the polymerization process. Figure 6 presents different routes in the reaction of oxygen with the ZN catalyst.

Figure 8 depicts the interaction of oxygen with the ZN catalyst. In this pathway, an oxygen molecule coordinates with the central metal, titanium, utilizing the two available vacant orbitals. This coordination results in the formation of a peroxide bond (a), characterized by one oxygen atom that is bound to another oxygen atom by a single bond. In this configuration, the coordinated oxygen molecule can subsequently bond with another titanium metal center, maintaining the peroxide nature of the bond (b). Furthermore, there is the possibility of two diatomic oxygen molecules coordinating with two metal centers, also forming a peroxide bond between the metal centers and the oxygen atoms (c). This type of peroxide bond has been extensively studied and observed in systems involving transition metals, such as copper [44]. The presence of these peroxide bonds influences the properties and reactivity of organometallic compounds, as they can participate in various chemical reactions and play a fundamental role in catalysis. The detailed analysis of the reaction pathways presented in Figure 8 provides a deeper understanding of the reactive mechanisms and intermediate species in catalytic systems. This knowledge is essential for the design and development of more efficient strategies in the synthesis of organic and organometallic compounds and the controlled manipulation of catalytic reactions.

### 3.3. PP Thermal Analysis

#### Thermal Degradation of the Material

Sample degradation analyses were performed using the Thermogravimetric Analysis (TGA) technique, and the results are shown in Figure 9. The DSC data show that as the temperature increases, the three gases (oxygen, carbon dioxide, and carbon monoxide) experience a decrease in their concentrations in the system. This suggests that these gases are released from the system at higher temperatures. Furthermore, at a higher temperature of 470 °C, a stabilization of the concentrations of the gases at low levels is observed, suggesting that they are no longer evolved or released in significant quantities at higher temperatures. The inflection points of decrease indicate the temperatures at which the gases (O_2_, CO_2_, and CO) are removed or released from the material. These temperatures are 390, 380, and 380 °C for O_2_, CO_2_, and CO, respectively. As the concentrations of these gases decrease with the increasing temperature, they likely contribute to the degradation of PP. The release of these gases may be related to the thermal decomposition reactions of the polymer, such as the degradation of polymer chains or the release of volatile decomposition products [45,46].

High concentrations of CO_2_ and CO may be associated with PP decomposition reactions since these gases may indicate the release of decomposition products. An increase in the CO_2_ concentration can also indicate the presence of polymer oxidation reactions. On the other hand, oxygen can inhibit some thermal degradation reactions since it can be involved in oxidation processes. However, if volatile decomposition products are formed, oxygen can contribute to their release and thus to the degradation of PP.

In Dtga (Figure 10), it was observed that the PP with CO_2_ obtained higher percentages of weight loss compared to the cases of CO and O_2_ around 325 °C. This suggests that the presence of CO_2_ accelerated the thermal degradation of PP at that temperature.

It was observed that, at a temperature of 450 °C, the PP with CO experienced the most significant weight loss, reaching 23.06%, while for CO_2_ at the same temperature, the weight loss was 19.7%, and for O_2_, the most significant weight loss occurred at 440 °C with 19.7%. These results indicate that PP with CO undergoes more significant thermal degradation than CO_2_ and O_2_ at that high temperature.

## 4. Conclusions

In this study, we investigated the interactions of CO_2_, CO, and O_2_ with the Ti active center on the MgCl_2_ surface, focusing on their potential effects on the catalytic reaction. We used a different model of the MgCl2 surface, specifically the Mg_9_Cl_18_ cluster obtained from relaxed MgCl2 surfaces, and examined the adsorption of these toxic molecules on the laterally cut (110) surface of MgCl2. It was observed that carbon monoxide (CO) exhibited the most negative *Ead*, indicating substantially stronger adsorption compared to other molecules such as molecular oxygen (O_2_) and carbon dioxide (CO_2_). This finding suggests a preference of the catalyst for CO due to its strong interaction. This strong binding could lead to a greater inhibition of the catalytic activity of ZN, as CO firmly attaches to the catalyst, preventing it from participating in polymerization reactions.

During the experimental tests, it was observed that CO in the catalytic system resulted in a significant decrease in polymer productivity. At low CO concentrations, there was a 15% reduction in productivity, while at higher concentrations, this reduction reached up to 40%. Similarly, CO caused a reduction in the Polymer Flow Index (MFI), with a decrease of 20% compared to the CO-free system. This indicates lower processability and fluidity of the formed polymer. Furthermore, the presence of CO significantly affected the molecular weight (Mw) of the produced polymer. A 10% decrease in the average molecular weight of the polymer was observed compared to the CO-free conditions. This could be due to the incorporation of CO into the polymer structure or its effect as a modifier in the growth of polymer chains. Thermal degradation analyses and global descriptor values also supported our experimental findings. The adsorption energy of CO on the active center of the catalyst was more favorable compared to CO_2_, suggesting a higher affinity and poisoning capacity of the catalyst by CO. Additionally, the global descriptors revealed that CO exhibited higher reactivity compared to CO_2_ and O_2_, supporting its role as the most potent inhibitor of the polymerization process.

## Figures and Tables

**Figure 1 polymers-16-00605-f001:**
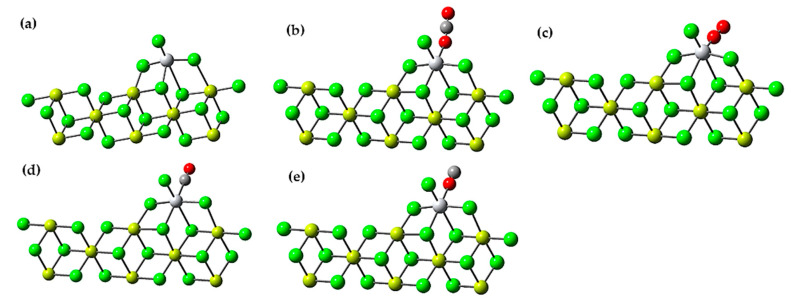
(**a**) Ziegler-Natta catalyst cluster Mg_8_Cl_16_; (**b**) Adsorption of CO_2_ on the active center of the Ziegler-Natta catalyst; (**c**) Adsorption of O_2_ on the active center of the Ziegler-Natta catalyst; (**d**) Adsorption of CO by the carbon atom on the active center of the Ziegler-Natta catalyst; (**e**) Adsorption of CO by the oxygen atom on the active center of the Ziegler-Natta catalyst. Chlorine atoms are indicated in green, magnesium atoms are in yellow, oxygen atoms in red, carbon atoms in dark gray, and titanium in gray.

**Figure 2 polymers-16-00605-f002:**
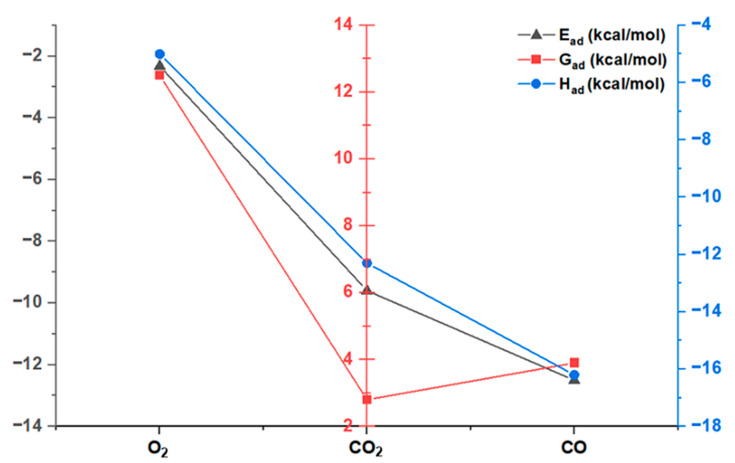
Relationship between adsorption energy, Gibbs free energy of adsorption, and enthalpy of adsorption of the inhibitors.

**Figure 3 polymers-16-00605-f003:**
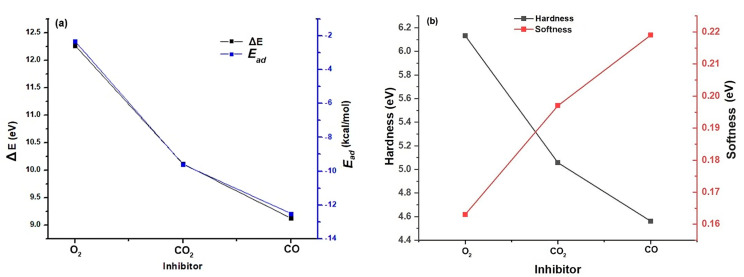
(**a**) Relationship between adsorption energy and energy gap for each inhibitor. (**b**) Relationship between global softness and hardness for each inhibitor.

**Figure 4 polymers-16-00605-f004:**
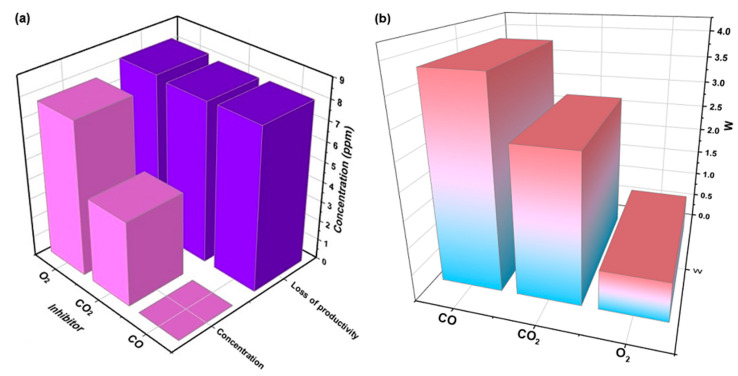
(**a**) General relationship between the productivity loss of the Ziegler–Natta catalyst as a function of the concentration of each inhibitor. (**b**) Electrophilicity index of each of the inhibitors.

**Figure 5 polymers-16-00605-f005:**
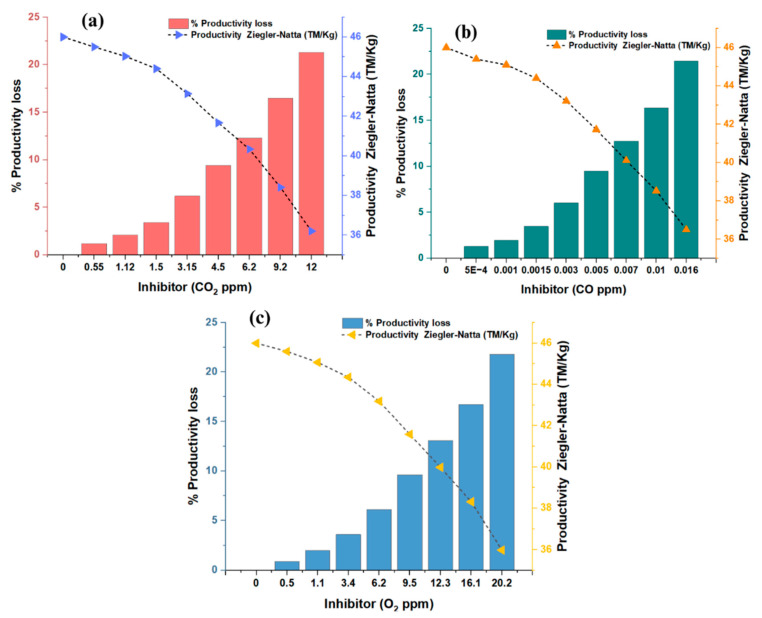
(**a**) Effects of CO_2_ on catalyst productivity and lost productivity. (**b**) Effects of CO on catalyst productivity and lost productivity. (**c**) Effects of O_2_ on catalyst productivity and lost productivity.

**Figure 6 polymers-16-00605-f006:**
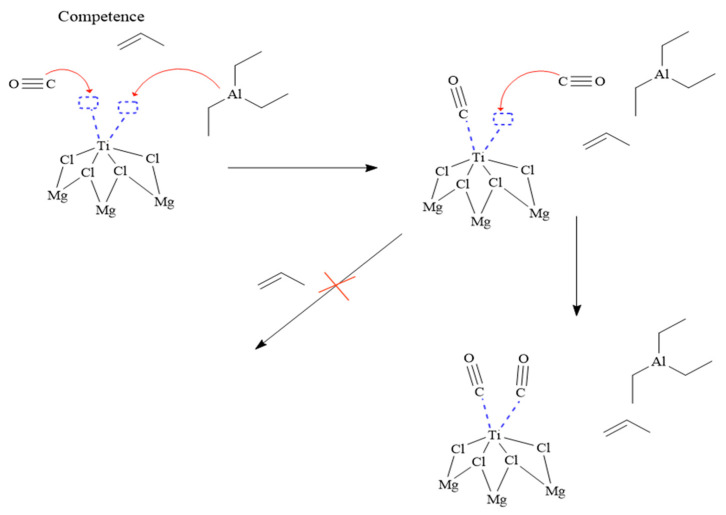
Reaction mechanism of carbon monoxide with the ZN catalyst.

**Figure 7 polymers-16-00605-f007:**
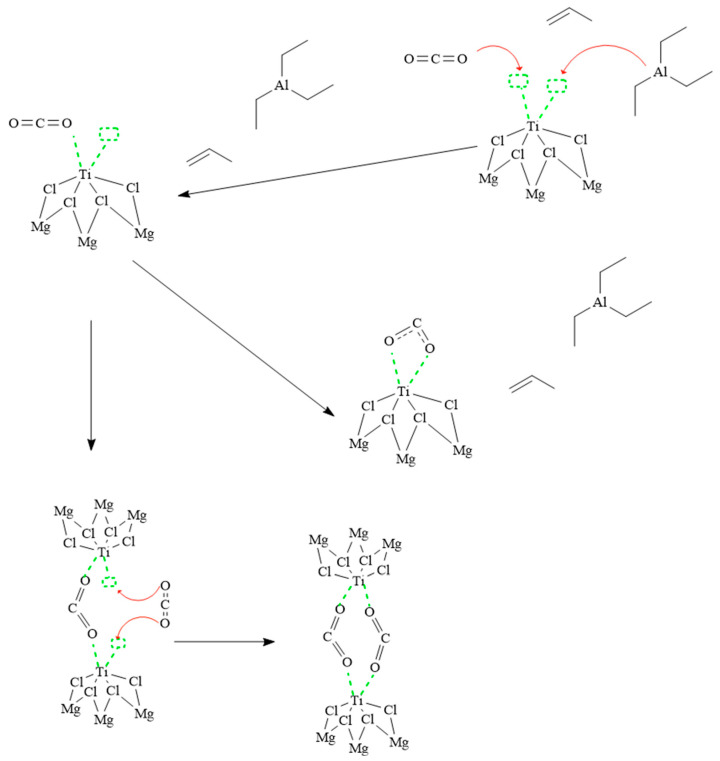
Reaction mechanism of carbon dioxide with the ZN catalyst.

**Figure 8 polymers-16-00605-f008:**
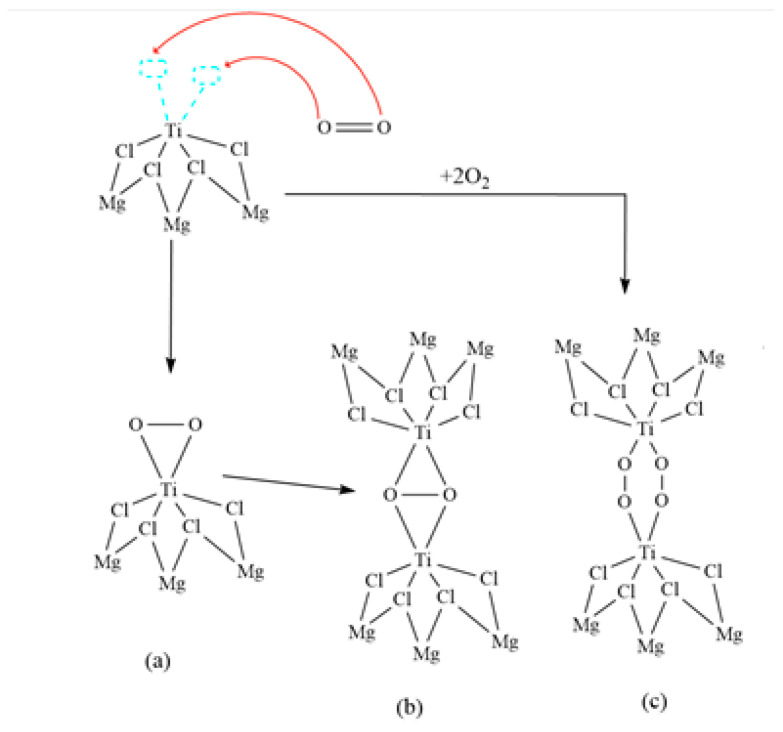
Reaction mechanism of oxygen with the ZN catalyst; (**a**) O-O bond (peroxo); (**b**) a coordinated oxygen molecule with two metal centers; (**c**) two diatomic oxygen molecules coordinated with two metal centers.

**Figure 9 polymers-16-00605-f009:**
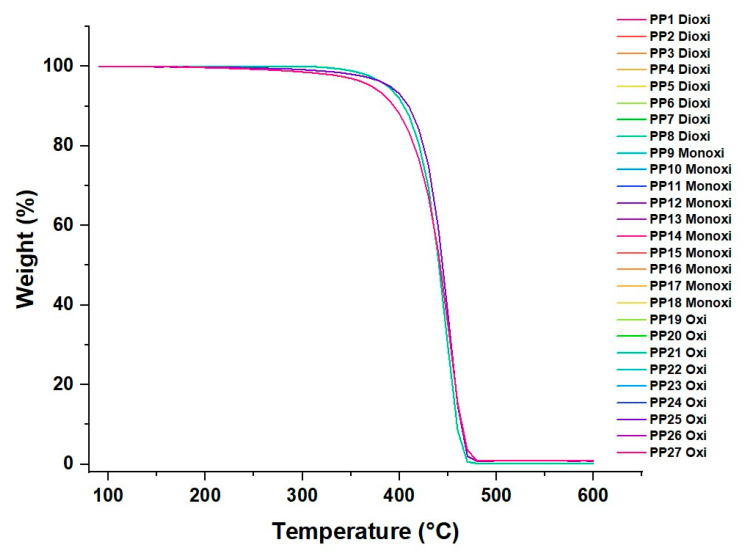
Polypropylene TGA with traces of CO_2_, CO, and O_2_.

**Figure 10 polymers-16-00605-f010:**
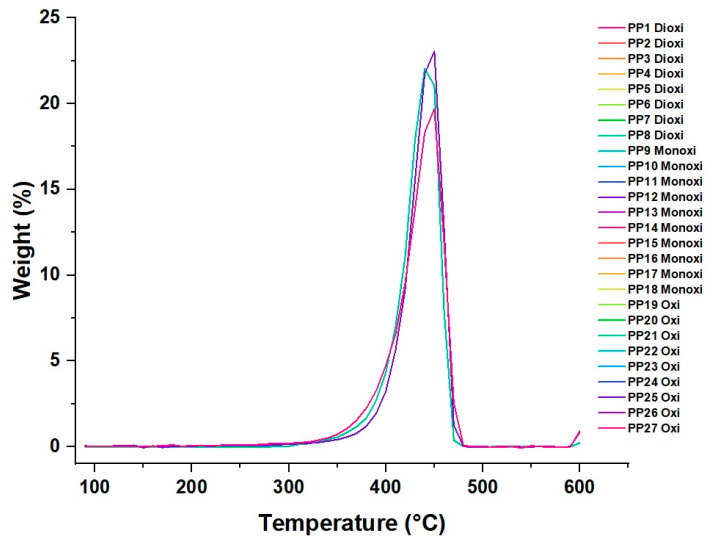
Polypropylene dTGA with amounts of CO_2_, CO, and O_2_.

**Table 1 polymers-16-00605-t001:** CO_2_, CO, and O_2_ concentrations collected during the polymerization process.

Materials	Catalyst Kh/h	TEAl Kg/h	CO_2_ (ppm)	CO (ppm)	O_2_ (ppm)	T °C	Pressure Bar
PP1	5	0.25	0	0	0	70	27
PP2	5	0.25	0.55	0	0	70	27
PP3	5	0.25	1.12	0	0	70	27
PP4	5	0.25	1.5	0	0	70	27
PP5	5	0.25	3.15	0	0	70	27
PP6	5	0.25	4.5	0	0	70	27
PP7	5	0.25	6.2	0	0	70	27
PP8	5	0.25	9.2	0	0	70	27
PP9	5	0.25	12	0	0	70	27
PP10	5	0.25	0	0	0	70	27
PP11	5	0.25	0	0.0005	0	70	27
PP12	5	0.25	0	0.001	0	70	27
PP13	5	0.25	0	0.0015	0	70	27
PP14	5	0.25	0	0.003	0	70	27
PP15	5	0.25	0	0.005	0	70	27
PP16	5	0.25	0	0.007	0	70	27
PP17	5	0.25	0	0.01	0	70	27
PP18	5	0.25	0	0.016	0	70	27
PP19	5	0.25	0	0	0	70	27
PP20	5	0.25	0	0	0.5	70	27
PP21	5	0.25	0	0	1.1	70	27
PP22	5	0.25	0	0	3.4	70	27
PP23	5	0.25	0	0	6.2	70	27
PP24	5	0.25	0	0	9.5	70	27
PP25	5	0.25	0	0	12.3	70	27
PP26	5	0.25	0	0	16.1	70	27
PP27	5	0.25	0	0	20.2	70	27

**Table 2 polymers-16-00605-t002:** *E_ad_*, *G_ad_*, and *H_ad_* values in kcal/mol for the interaction of the three inhibitors with the active titanium center of the Ziegler–Natta catalyst.

Type of Poison	*E_ad_*	*G_ad_*	*H_ad_*
O_2_	−2.32	12.5	−5.0
CO_2_	−9.6	2.8	−12.3
CO	−12.5	3.9	−16.2

**Table 3 polymers-16-00605-t003:** Global descriptors for CO_2_, O_2_, and CO.

Parameter (eV)	O_2_	CO_2_	CO
η	6.1336	5.0569	4.5620
*S*	0.163	0.197	0.219
µ	−3.008	−5.248	−5.889
ω	0.737	2.723	3.800

**Table 4 polymers-16-00605-t004:** Local descriptors of CO_2_, CO, and O_2_.

Inhibitor	Atom	$f^{-}$	$f^{+}$	$f^{0}$	Δf
O_2_	O	0.500	0.500	0.500	0
	O	0.500	0.500	0.500	0
CO	C	0.6717	0.6977	0.6847	0.026
	O	0.3283	0.3023	0.3153	−0.026
CO_2_	C	0.0027	0.5661	0.2844	0.5634
	O	0.4986	0.2169	0.3578	−0.2817
	O	0.4986	0.2169	0.3578	−0.2817

## Data Availability

Data are contained within the article.

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
