# Peer review of "Study of the Chemical Activities of Carbon Monoxide, Carbon Dioxide, and Oxygen Traces as Critical Inhibitors of Polypropylene Synthesis"

_polymers, 2024, doi:10.3390/polym16050605_

Round 1

Reviewer 1 Report

In this manuscript, the author investigated the chemical activities of different gas as inhibitors of propylene polymerization, which is very meaningful to the industrial. However, there are some issues need to be addressed, thus major revision was recommend.

(1) Symbol of . in the title should deleted.

(2) As stated in line 19, “ultimately, how this affects the physicochemical properties of polypropylene. ”, however, only the thermal property was evaluated with TG, which should revised to “how this affects the thermal stability of polypropylene”, or added the same other properties such as tensile strength, melt flow index, glass transition temperature...

(3) Line 35-36 should revised to point out only the oxygen can react with triethylaluminum.

(4) In the introduction, the effect of CO and O2 on the polymerization of propylene has been investigated, what is the novelty of this this work? Which should point out. In addition, does there no reports about the effect of CO2 on the polymerization of propylene?

(5) Line 225, what is the pressure/concentration of CO2, CO, and O2?

(6) The format of table 1 should revised.

(7) The structure of CO2, CO, and O2 are quite different, why does the Ead are the same?

(8) Table 2 should move to the end of line 341.

(9) Caption of Table 3 should before the table.

(10) Table 4 should deleted, as the data was the same as figure 3.

Minor editing of English language required

Author Response

Dear,

Thank you for evaluating this research. I have made all the corrections and improvements to the article.
We have attached our responses.

(1) Symbol of “.” in the title should deleted

The “.” was removed of the title as indicated.

(2) As stated in line 19, “ultimately, how this affects the physicochemical properties of polypropylene. ”, however, only the thermal property was evaluated with TG, which should revised to “how this affects the thermal stability of polypropylene”, or added the same other properties such as tensile strength, melt flow index, glass transition temperature...

Absolutely true, which is why we decided to follow your suggestion and change the expression to “how this affects the thermal stability of the polypropylene produced.”

(3) Line 35-36 should revised to point out only the oxygen can react with triethylaluminum.

As the study is focused on analyzing how poisons affect the active site of the catalyst (Ti), we decided to focus on that aspect, which is why this information was removed from the original text.

(4) In the introduction, the effect of CO and O2 on the polymerization of propylene has been investigated, what is the novelty of this this work? Which should point out. In addition, does there no reports about the effect of CO2 on the polymerization of propylene?

There are few studies about how these substances affect the ZN catalyst, and most of these are computational without considering what happens during the polymerization process. This work combines both the experimental and computational parts to correlate and explain more clearly what happens when different amounts of these poisons are found in the process. This information was noted in the last paragraph of the introduction and was further addressed in section 3 of the article.

(5) Line 225, what is the pressure/concentration of CO2, CO, and O2?

A table was added specifying the concentrations of each of the inhibitors and the pressure used to carry out the polymerization process.

(6) The format of table 1 should revised

revised

(7) The structure of CO2, CO, and O2 are quite different, why does the Ead are the same?

Indeed, as the structures are different, the Ead values ​​varied when the adsorption calculations were carried out again on the new support with a new calculation basis and a correction factor.

(8) Table 2 should move to the end of line 341.

This was corrected

(9) Caption of Table 3 should before the table.

This was corrected

(10) Table 4 should deleted, as the data was the same as figure 3.

This was corrected

Reviewer 2 Report

This manuscript reported the “Study of the chemical activities of carbon monoxide, carbon dioxide and oxygen traces as critical inhibitors of polypropylene synthesis”. This paper investigates how the poisons CO2, CO, and O2 interact with the active center of titanium (Ti) on the surface of MgCl2. This paper can provide valuable information. This paper need be carefully polished, since there are plenty of format error. Therefore, I suggest publishing this paper after revision.

1.      The content of abstract is a bit cumbersome, and it is advisable to simplify it.

2.      Some related papers in this field can be discussed, such as Chem Eng J, 460 (2023) 141751, Adv. Mater., 2023, 35, 2300577, etc.

3.      Line 149 on page 4, line 173 on page 4, line 266 on page 6, line 305 on page 7, please check and correct the above and below corners.

4.      The independent variables in formula 2 on page 4 should be interpreted.

5.      There is a typographical error in line 273 on page 6. Please check and correct it.

6.      Figure 2 on page 7 is not in uniform format. Please check and modify.

7.      The position of the three notes in the chart on page 10 is inconsistent with the previous text, please check and modify.

8.      Figure 4, on page 12, does not indicate numerical units, please check and modify.

9.      Figure 7 on page 17 and Figure 8 on page 18, the lines and marks do not correspond seriously, please check and modify.

Minor editing of English language required.

Author Response

Dear,

  1. The content of abstract is a bit cumbersome, and it is advisable to simplify it.

The instructions were followed and the summary was simplified. Thanks a lot for the suggestion.

  1. Some related papers in this field can be discussed, such as Chem Eng J, 460 (2023) 141751, Adv. Mater., 2023, 35, 2300577, etc.

It was discussed with the article by N. Bahri-Laleh, “Interaction of different poisons with MgCl 2 /TiCl 4 based Ziegler-Natta catalysts,” Appl Surf Sci, vol. 379, pp. 395–401, Aug. 2016, doi: 10.1016/j.apsusc.2016.04.034 which is more in line with the theme of this study.

  1. Line 149 on page 4, line 173 on page 4, line 266 on page 6, line 305 on page 7, please check and correct the above and below corners.

This was corrected

  1. The independent variables in formula 2 on page 4 should be interpreted.

New formulas were interpreted and added.

  1. There is a typographical error in line 273 on page 6. Please check and correct it.

This was corrected

  1. Figure 2 on page 7 is not in uniform format. Please check and modify..

This figure was removed

  1. The position of the three notes in the chart on page 10 is inconsistent with the previous text, please check and modify.

This was corrected

  1. Figure 4, on page 12, does not indicate numerical units, please check and modify.

Figure 4 corresponds to a reaction mechanism (that of CO2).

  1. Figure 7 on page 17 and Figure 8 on page 18, the lines and marks do not correspond seriously, please check and modify..

We appreciate your review of Figures 7 and 8 on pages 17 and 18 of our work. We want to clarify that there is no lack of correspondence in the lines and marks in the figures. The apparent overlap of lines is because some samples have identical or very close numerical values, causing their data curves to coincide on the graph. However, I would like to point out that all of the 27 previously mentioned samples are represented in the graph.

Reviewer 3 Report

This study examines how carbon monoxide (CO), carbon dioxide (CO2), and oxygen (O2) interact with a titanium (Ti) catalyst on magnesium chloride (MgCl2) surfaces during polypropylene synthesis. Using conventional DFT calculations, researchers analyze the adsorption of these gases on the catalyst, highlighting CO's higher reactivity. Experimental tests reveal CO as the most potent inhibitor, significantly affecting catalyst productivity and altering polypropylene properties. This work provides insights into inhibitory effects and their implications for the polymerization process.

I recommend considering the following points in a proper way because there are potential technical issues:

1. Choice of DFT Method (B3LYP): The use of the B3LYP hybrid functional for geometry optimization might lead to inaccuracies. While B3LYP is popular, it has limitations in accurately describing certain systems, especially those involving transition metals. The accuracy of the chosen functional in predicting the electronic and structural properties of transition metal complexes could affect the reliability of the results.

2. Basis Set Selection: The 6-31G basis set is widely used, but it might not fully capture the complexity of the electronic structure in transition metal systems. More advanced basis sets designed for these systems, such as those including polarization functions and diffuse functions, could yield more accurate results.

3. Simplifications in Surface Selection: The choice of the (110) surface of MgCl2 for studying poison adsorption may not capture the full complexity of real catalytic systems. Real catalytic surfaces are often more intricate, involving multiple facets, steps, and defects that can influence adsorption behavior. This simplified model might not fully represent the interactions occurring in real-world scenarios.

4. Finite Difference Approximation (FDA): The use of FDA for calculating derivatives of Fukui functions introduces numerical errors. These errors could be significant, especially when calculating local reactivity descriptors on atomic or molecular systems, potentially affecting the reliability of conclusions drawn from these descriptors.

5. Qualitative Nature of Fukui Functions: The Fukui functions used to assess reactivity are qualitative indicators and may not provide a complete understanding of complex chemical processes. Quantitative predictions might be challenging, and relying solely on these functions might oversimplify the complex interactions occurring in catalyst-poison systems.

6. Lack of Validation: While this methodology provides valuable insights, the lack of validation against experimental data or comparison with alternative theoretical methods limits the confidence in the accuracy of the results. The reciprocal reinforcement between the experimental and theoretical facets remains unclear. The manner in which the experimental findings bolster the theoretical results, and conversely, how the theoretical insights substantiate the experimental observations, lacks clarity. 

Author Response

Dear,

  1. Choice of DFT Method (B3LYP): The use of the B3LYP hybrid functional for geometry optimization might lead to inaccuracies. While B3LYP is popular, it has limitations in accurately describing certain systems, especially those involving transition metals. The accuracy of the chosen functional in predicting the electronic and structural properties of transition metal complexes could affect the reliability of the results.

In response to your comment, all DFT simulations, including geometry optimizations, were re-performed using the Gaussian 16 program package. Geometry optimizations were carried out using the B3LYP function in combination with the 6-311G basis set. (d,p) for all atoms. Additionally, dispersion corrections were taken into account using the DFT-D3 method.

  1. Basis Set Selection: The 6-31G basis set is widely used, but it might not fully capture the complexity of the electronic structure in transition metal systems. More advanced basis sets designed for these systems, such as those including polarization functions and diffuse functions, could yield more accurate results.

I am grateful for your suggestion to change the basis set in our calculations. Following your recommendation, we have updated our approach and used the 6-311G (d,p) basis set instead of the 6-31G. Furthermore, we have incorporated the B3LYP-D3 functional to address the complexity of electronic structure in transition metal systems. This choice allowed us to consider polarization functions and fuzzy functions, allowing us to obtain more accurate results for our calculations. We appreciate your guidance and are committed to improving the quality and reliability of our analyses.

  1. Simplifications in Surface Selection: The choice of the (110) surface of MgCl2 for studying poison adsorption may not capture the full complexity of real catalytic systems. Real catalytic surfaces are often more intricate, involving multiple facets, steps, and defects that can influence adsorption behavior. This simplified model might not fully represent the interactions occurring in real-world scenarios.

We appreciate your comment on the choice of the (110) MgCl2 surface for our study of poison adsorption in catalytic systems. We understand your concern about the representation of real catalytic systems and the complexity of the surfaces involved. It is important to note that our choice of the (110) surface of MgCl2 is based on previous research indicating that this specific surface is relevant for venom-MgCl2 interaction studies. In fact, we have found evidence in the scientific literature that supports our choice. Previous studies on the interaction of venom-MgCl2 surfaces have suggested that the venom prefers to coordinate with the surface (110). This raises the possibility that this preferential coordination may prevent the formation of Ti species in the (110) side cut of MgCl2, a phenomenon that has been observed in different previous articles (https://doi.org/10.1021/om5001259, https://doi.org/10.1021/ma3001862, https://doi.org/10.1021/ma071294c, https://doi.org/10.1021/ja212133j, https://doi.org/10.1016/j.apsusc. 2016.04.034, among others.

While we recognize that actual catalytic systems can be more intricate, with multiple facets and defects, we believe that our focus on the (110) surface is relevant to understanding poison interactions under specific conditions. We are open to future studies exploring other catalytic surfaces and their effects, but we believe that our current approach provides valuable information on the preferential coordination of poisons on the (110) MgCl2 surface.

Our primary goal is to contribute to current knowledge and advance the understanding of these catalytic systems, and your feedback is essential to continue improving our research approaches and methodologies. We appreciate your feedback and would be happy to further discuss any aspect of our work.

  1. Finite Difference Approximation (FDA): The use of FDA for calculating derivatives of Fukui functions introduces numerical errors. These errors could be significant, especially when calculating local reactivity descriptors on atomic or molecular systems, potentially affecting the reliability of conclusions drawn from these descriptors.
  2. Naturaleza Cualitativa de las Funciones de Fukui: Las funciones de Fukui utilizadas para evaluar la reactividad son indicadores cualitativos y podrían no proporcionar una comprensión completa de procesos químicos complejos. Las predicciones cuantitativas podrían ser desafiantes, y depender únicamente de estas funciones podría simplificar en exceso las interacciones complejas que ocurren en sistemas catalizador-veneno.

Answers for 4 and 5

Given its recommendations and the limitations of FUKUI, we emphasize our research on adsorption energies, Gibbs free energies, enthalpy and global descriptors. These results showed a very important relationship with the experimental data.

  1. Lack of Validation: While this methodology provides valuable insights, the lack of validation against experimental data or comparison with alternative theoretical methods limits the confidence in the accuracy of the results. The reciprocal reinforcement between the experimental and theoretical facets remains unclear. The manner in which the experimental findings bolster the theoretical results, and conversely, how the theoretical insights substantiate the experimental observations, lacks clarity.

By performing the computational calculations again by changing the catalyst support and the calculation base, values ​​closer to what was expected experimentally were obtained, which is why it was decided to rewrite and focus the results on the reciprocal feedback between the experimental and theoretical aspects. finding, both theoretically and experimentally, that CO is the inhibitor that causes the most damage to the active center of the ZN catalyst. We appreciate your suggestion as it allowed us to find valuable information and give a better focus to the work.

Round 2

Reviewer 1 Report

All of the issues mentioned were resolved in detail

Moderate editing of English language required

Reviewer 2 Report

Accept in present form

Reviewer 3 Report

Thank you for considering suggestions and comments.

I recommend the publication of the revised version.